# Targeting OPA1-Mediated Mitochondrial Fusion Contributed to Celastrol’s Anti-Tumor Angiogenesis Effect

**DOI:** 10.3390/pharmaceutics15010048

**Published:** 2022-12-23

**Authors:** Gaofu Li, Lei Zhou, Huifang Deng, Congshu Huang, Ningning Wang, Lanxin Yue, Pengfei Zhang, Yongqiang Zhou, Wei Zhou, Yue Gao

**Affiliations:** 1Department of Pharmaceutical Sciences, Beijing Institute of Radiation Medicine, Beijing 100850, China; 2School of Pharmacy, Guangdong Pharmaceutical University, Guangzhou 510006, China; 3School of Chinese Materia Medica, Tianjin University of Traditional Chinese Medicine, Tianjin 301617, China

**Keywords:** celastrol, tumor angiogenesis, mitochondrion-targeted medicine, OPA1

## Abstract

Celastrol, an active triterpenoid extracted from one of the most famous traditional Chinese medicines (TCMs), *Tripterygium wilfordii* Hook.f., is a novel anti-cancer drug with significant anti-angiogenesis activity. However, the exact molecular mechanisms underlying its anti-tumor angiogenesis effect remain unclear. The process of angiogenesis needs lots of energy supply, which mostly derives from mitochondria, the “energy factory” in our body. This study shows that celastrol exerts visible suppression on tumor growth and angiogenesis in a cell-derived xenograft (CDX). Likewise, it reduced the tube formation and migration of human umbilical vein endothelial cells (HUVECs), suppressed the energy metabolism of mitochondria in the Seahorse XF Mito Stress Test, and triggered mitochondrial fragmentation and NF-κB activation. Mechanically, celastrol downregulated the expression of mitochondrial-sharping protein optic atrophy protein 1 (OPA1), which was further estimated by the OPA1 knockdown model of HUVECs. Specifically, celastrol directly suppressed OPA1 at the mRNA level by inhibiting the phosphorylation of STAT3, and stattic (STAT3 inhibitor) showed the same effects on OPA1 suppression and anti-angiogenesis activity. Overall, this study indicates that celastrol inhibits tumor angiogenesis by suppressing mitochondrial function and morphology via the STAT3/OPA1/P65 pathway and provides new insight for mitochondrion-targeted cancer therapy.

## 1. Introduction

*Tripterygium wilfordii* Hook.f. is a famous traditional Chinese medicine (TCM), broadly distributed in East Asia, and has been used for more than 2000 years. One of its most promising bioactive components is celastrol, a pentacyclic triterpenoid with multi-targeting potential in anti-tumor therapy, such as promoting tumor cell apoptosis by accumulating reactive oxygen species (ROS), decreasing mitochondrial membrane potential, upregulating pre-apoptosis proteins [1], and suppressing tumor growth by triggering lipophagy in clear cell renal cell carcinoma (ccRCC) [2]. However, the precise molecular mechanisms underlying its anti-tumor effect are less well understood. Recently, several studies reported the obvious inhibitory effect of celastrol on the proliferation, migration, and invasion of endothelial cells such as HUVECs, which makes it possible to elucidate its anti-tumor effect from the perspective of inhibiting the growth of blood vessels, namely anti-angiogenesis [3,4,5].

Angiogenesis is crucial for organ growth in the embryo and wounded tissue repair in adults [6], unfortunately, this process can also be activated in tumor genesis [7]. Because of partial hypoxia and lack of multi-nutritious inside the solid tumor with a diameter of more than 2–3 mm, the cancer cells secrete angiogenic factors, such as vascular endothelial growth factor (VEGF) and fibroblast growth factor (FGF), to generate new capillaries from pre-existing blood vessels mainly via the HIF–VEGF pathway [6,8,9], thus maintaining the oxygen and nutrition supply and metabolite discharge to fit the extensive proliferation of themselves. Angiogenesis does not cause malignant tumors but promotes tumor progression and metastasis. Prospectively, the endothelial cells (ECs) have stable genomics compared with tumor cells, hence they are less likely to show resistance in the therapy of anti-angiogenesis agents. Recently, cumulative evidence has verified that celastrol shows an evident anti-angiogenic effect and reverses the angiogenic effect of VEGF [4,10], meanwhile, it also downregulates the expression of HIF-1α and VEGF [3].

Mitochondria are organelles with a bilayer membrane, their division and fusion are crucial for their function. Mitofusins (MFN) 1 and 2, located in the outer mitochondrial membrane (OMM), regulate mitochondrial fusion [11], coordinating with optic atrophy protein 1 (OPA1), located in the inner mitochondrial membrane (IMM), which promotes the fusion of IMM [12,13]. OPA1 belongs to the dynamin family of large GTPases, which can be cleavage by the mitochondrial processing peptidase (MMP) into eight kinds of messenger RNA (mRNA) to generate long isoforms, L-OPA1 [12]. Furthermore, L-OPA1 can be processed by YMEL and OMA1 in the N-terminal region to the GTPase domain and generate short isoforms, S-OPA1 [13,14]. Both L- and S-OPA1 are crucial for IMM fusion [15]. Recent studies tied angiogenesis and OPA1 closely [16,17,18], confirming that the upregulation of OPA1 promotes angiogenesis by limiting NF-κB signaling. This finding provides a promising prospect for anti-tumor angiogenesis therapy. Angiogenesis is an energy-consuming process [19], these studies focus on mitochondria, the energy factory in cells, and achieved positive results.

In this study, we show that celastrol can significantly inhibit angiogenesis inside the solid tumor, at the same time, the mitochondrial length of HUVECs is dramatically decreased. Thus, we speculate that energy supply capacity change caused by mitochondrial morphology alteration may be one of the reasons that celastrol suppresses angiogenesis.

## 2. Materials and Methods

### 2.1. Cell Lines

Non-small cell lung cancer (NSCLC) cell line A549 and the HUVEC cell line were kindly provided by Prof. Zhou Wei at the Beijing Institute of Radiation Medicine. A549 cells were cultured in Dulbecco’s modified Eagle medium (DMEM) (Gibco, Thermo Fisher Scientific, Waltham, MA, USA) supplemented with 10% fetal bovine serum (FBS) (Gibco) and 1% penicillin/streptomycin (Hyclone, Logan, UT, USA). HUVECs were cultured in DMEM (no sodium pyruvate) (Gibco) supplemented with 10% fetal bovine serum (FBS) (Gibco) and 1% penicillin/streptomycin (Hyclone), or endothelial cell growth medium-2 (EGM-2) which was supplemented with VEGF (Lonza, Switzerland). All the cells were cultured at 37 °C and 5% CO_2_ in an incubator.

### 2.2. Animals and Treatments

The cell-derived xenograft (CDX) model was constructed by subcutaneous injection of the 1:1 (*v*:*v*) mixture of A549 cells (1 × 10^6^ cells per mouse) and matrix gel (Corning, Corning, NY, USA) at the scapula after 7 days of acclimatization of 10 BABL/c Nude mice. Ten days later, the transplanted tumors were formed and the animals were randomly divided equally into control and celastrol administration groups (i.p., 2 mg/g/day), then kept feeding for 20 days. Mice were housed in colony cages in 12 h light/12 h dark cycles. BABL/c Nude mice (4-week-old, ~18 g) were purchased from SPF Laboratories (Beijing Vital River, Beijing, China, certificate No. SCXK2016-0011). All animal experiments were approved by the Ethics Committee of the Beijing Institute of Radiation Medicine (Approval No. IACUC-DWZX-2022-822).

### 2.3. Hematoxylin and Eosin Staining and Immunofluorescence

Tumor tissues were fixed in 10% paraformaldehyde solution (Servicebio, Wuhan, China) over 7 h, then embedded in paraffin (Servicebio, China). According to the standard protocol, hematoxylin and eosin (H&E) staining and immunofluorescence (IF) were carried out on 5 µm tissue slices. Antibodies used for IF included Rabbit Anti-CD31 Polyclonal antibody (proteintech, Wuhan, China, cat# 11265-1-AP) and Purified Mouse Anti-OPA1 antibody (BD Biosciences, Franklin Lakes, NJ, USA, cat# 612606).

### 2.4. Cell Viability Assay

Cell viability was determined following the instruction of the cell counting kit-8 assay (CCK-8, Dojindo, Japan). Five thousand HUVECs were seeded in a 96-well plate per well, after 24 h, they were treated with celastrol (0, 15.63, 31.25, 62.50, 125.0, 250.0, 500.0, 1000, 2000, and 4000 nM) or 250 nM celastrol with stattic (0, 125.0, 250.0, 500.0, 1000, 2000, and 4000 nM) in EGM-2 for 24 h. Then cells were washed with PBS buffer (Gibco) and incubated with DMEM containing 10% CCK-8 solution for 2 h in an incubator with 5% CO_2_ at 37 °C. After that, the absorbance of each well was detected with a microplate reader at 450 nm (Multiskan MK3, Thermo Fisher Scientific, USA). Cell viabilities for celastrol groups were presented as percentages of that of the control.

### 2.5. Matrigel Tube Formation Assay

Briefly, the inner wells of an angiogenesis slide (ibidi, Germany, cat# 81506) were filled with 10 µL growth factor reduced Matrigel (BD Biosciences, cat# 356231), after polymerizing for 30 min in a 37 °C incubator, 50 µL HUVEC suspensions (1000 HUVECs) with or without 250 nM celastrol in EGM-2 were applied into the upper well. The cells were incubated at 37 °C with 5% CO_2_ for 6 h, followed by 30 min incubation with 50 µL calcein (6.25 µg/mL, Thermo Fisher, USA) at room temperature (R.T.) and one wash with PBS buffer (Gibco). The macroscope of each well was detected with a cell imaging multimode reader (Cytation 7, BioTek, Winooski, VT, USA). Calcein was excited with the 494–514 nm LEDs. The nodes of tubes, number of tubes, and tube length were identified and analyzed via the plugin “angiogenesis analyzer” for image J.

### 2.6. Measurement of Mitochondrial Length by High-Content Screening

A total of 5000 HUVECs were seeded in a Cell Carrier 96-well ultra microplate (PerkinElmer, Waltham, MA, USA), after 24 h, they were treated with 250 nM celastrol or 500 nM stattic for 24 h. According to the standard protocol, the cells were washed with PBS buffer (Gibco) and incubated with DMEM containing 100 nM MitoTracker^®^ Deep Red FM (Thermo) at 37 °C with 5% CO_2_ for 30 min in an incubator. After rinsing with PBS, the cells were fixed in a 3.7% paraformaldehyde solution (Servicebio, China) at R.T. for 30 min, then covered with an antipode mounting medium containing DAPI. The detection and analysis were operated by the Operetta CLS High-Content Analysis System (PerkinElmer) with 63× Water/1.15 NA. MitoTracker^®^ Deep Red FM and DAPI were excited with the 630–660 and 355–385 nm LEDs, respectively.

### 2.7. Wound-Healing Migration Assay

Briefly, 70 µL HUVECs were seeded in culture-insert 2 wells (ibidi, cat# 80209) according to the manufacturer’s instructions, then incubated at 37 °C with 5% CO_2_ for 24 h. After that, the cells were treated with or without 250 nM celastrol in EGM2 for 24 h. Images of the cells were taken after 0, 4, 8, and 12 h of incubation with a cell imaging multimode reader (Cytation 7, BioTek, Winooski, VT, USA).

### 2.8. Measuring Oxygen Consumption Rate by Seahorse

The oxygen consumption rate (OCR) was analyzed by a Seahorse XF96 extracellular flux analyzer (Seahorse Bioscience, Agilent, TX, USA). HUVECs (10,000 cells per well) were seeded in XFe96 Cell Culture Microplates (101085-004, Agilent), then incubated at 37 °C with 5% CO_2_ for 24 h. After stabilization of the cellular metabolism, the cells were treated with or without 250 nM celastrol in EGM2 for 24 h. The subsequent experimental operations were performed in strict accordance with standard experimental procedures. Briefly, HUVECs were equilibrated in fresh XF DMEM base medium (103575-100, Agilent) containing 25 mM glucose (103577-100, Agilent), 1 mM sodium pyruvate (103578-100, Agilent), and 2 mM L-glutamine (103579-100, Agilent) for 1 h at a 37 °C incubator without extra CO_2_. For the Mito Stress test, 1.5 μM oligomycin A (Oligo), 4 μM carbonyl cyanide-p-trifluoromethoxy-phenylhydrazone (FCCP), and 0.5 μM rotenone/antimycin A (A/R) were sequentially injected into each well operated by the instrument.

### 2.9. Western Blot Assay

Firstly, following the manufacturer’s instructions, the concentrations of whole-cell proteins were qualified by abicinchoninic acid (BCA) protein assay kit (ApplygenTechnologies, Beijing, China). Then, the proteins were separated by sodium dodecyl sulfate-polyacrylamide gel electrophoresis (SDS-PAGE), transferred to PVDF membranes (IPVH00010, Millipore, Germany), and blocked in blocking buffer for 2 h at R.T. After that, the membranes were incubated with the following primary antibodies: Mitofusin-1 Rabbit Monoclonal antibody (Cell Signaling Technology, Danvers, MA, USA, cat#14739S), Mitofusin-2 Rabbit Monoclonal antibody (CST, cat#9482S), Purified Mouse Anti-OPA1 antibody (BD, cat# 612606), Anti-TOM20 Rabbit Monoclonal antibody (Abcam, Boston, MA, USA, cat#ab186735), COX IV Rabbit Monoclonal antibody (CST, cat#4850S), Phospho-NF-κB p65 (Ser536) Rabbit pAb (ZenBio, Chengdu, China, cat#340830), Stat3 Mouse Monoclonal antibody (CST, cat#9139S), Phospho-Stat3 (Tyr705) Rabbit Monoclonal antibody (CST, cat#9145S), Anti-GAPDH Rabbit Polyclonal antibody (Abcam, cat#ab9485), Anti-Beta Actin Recombinant antibody (Proteintech, cat# 81115-1-RR), and Anti-α-Tubulin antibody (CST, cat#2144S). Then, the membranes were washed and incubated with secondary antibodies (Goat Anti-Rabbit IgG H&L (HRP) (Abcam, cat#ab6721) or Goat Anti-Mouse IgG H&L (HRP) (Abcam, cat#ab6789)). The expression of proteins was detected by an enhanced chemiluminescence system (Millipore, Germany). Image J was used to measure the intensity of each band.

### 2.10. siRNA Transfection in HUVECs

siRNA targeting human OPA1 and non-specific control siRNA were obtained from HanBio (Shanghai, China). Cells were incubated in 6-well plates and transfected with different siRNA using Lipofectamine RNAiMAX Reagent (Thermo) according to the protocol. siRNA sequences were shown as below: forward, 5′-AGAAGAUGUUGAAAUGUAATT-3′; reverse, 5′-UUACAUUUCAACAUCUUCUTT-3′. After 48 h, the HUVECs in which OPA1 had been knocked down were identified and verified by Western blot.

### 2.11. RT-PCR and qPCR

Total cell RNA was prepared using TRIzol reagent (Thermo) following the manufacturer’s instructions. Total RNA was subjected to reverse transcription to synthesize cDNA using a One-Step gDNA Removal and cDNA Synthesis SuperMix (TransGen Biotech, Beijing, China). qPCR was performed using a PerfectStart Green qPCR SuperMix (TransGen Biotech). Genes-specific primers were used to amplify cDNA. The following primers were used:hs-OPA1: forward, 5′-CGACCCCAATTAAGGACATCC-3′;reverse, 5′-GCGAGGCTGGTAGCCATATTT-3′hs-GAPDH: forward, 5′-CTGGGCTACACTGAGCACC-3′;reverse, 5′-AAGTGGTCGTTGAGGGCAATG-3′

### 2.12. Transmission Electron Microscopy (TEM)

HUVECs were seeded in a 6-well microplate, and after 24 h, they were treated with or without 250 nM celastrol for 24 h. The cells were immobilized in 2.5% glutaraldehyde at 4 °C overnight then rinsed with precooled PBS and immobilized with 1% osmium tetroxide for 30 min at 4 °C. Next, the samples were dehydrated with gradient ethanol and embedded with epoxy resin. The embedded blocks were cut into ultra-thin sections of 50 nm using an ultra-thin slicer, fixed in a 200-mesh copper net, and stained with a uranium acetate staining solution. After drying, the ultrastructure of the cells was scanned by transmission electron microscopy (H-7650, Hitachi, Tokyo, Japan).

### 2.13. Computational Molecular Docking

The computational molecular docking study was carried out using Autodock Vina 1.2.3. The protein structure used in the docking studies was human STAT3, its PDB code is 6NJS. The compound used was celastrol, its CID code is 122724. The crystal structure was prepared according to the recommended procedure in the Protein Preparation Wizard. The ligands and solvent molecules were removed.

### 2.14. Statistical Analysis

All results were generated from at least three independent experiments, the statistics were presented as mean ± standard deviation (SD). One-way ANOVA followed by LSD post hoc test for multiple comparisons was performed for statistical analysis. The difference was considered statistically significant when *p* < 0.05.

## 3. Results

### 3.1. Celastrol Inhibits Lung-Derived Tumor Growth

We established a CDX model of A549 to investigate whether celastrol can suppress solid tumor growth or not. We took advantage of BABL/c Nude mice which have T cell deficiency due to lack of thymus, thus avoiding T cell influence in the evaluation of the anti-tumor effect. The experimental design is shown in Figure 1A. All the mice in the celastrol and control groups survived (Appendix A), celastrol predominantly suppressed tumor volume (Figure 1B), and the body weight of mice was slightly decreased in the celastrol group compared with the control group (Figure 1C and Appendix A). After 20 days of celastrol administration, the size and weight of solid tumors dramatically decreased (Figure 1D,E). These results indicated that celastrol can significantly inhibit solid tumor growth in vivo. We used H&E staining to further explore how celastrol shows the anti-tumor effect; the results illustrated that the nuclear size of a tumor cell has no significant deviation between the control and celastrol group (Figure 1F), but the number of vessels showed a remarkable reduction in the celastrol group (Figure 1F,G). This showed that celastrol may inhibit tumor growth by restraining tumor angiogenesis. Our results are consistent with previous studies of other tumor models [3,20].

### 3.2. Celastrol Inhibits HUVEC Tube Formation and Migration In Vitro

Next, we evaluated the angiogenesis capacity of HUVECs under celastrol treatment in vitro to further clarify our conclusion above. At first, we evaluated the cytotoxicity of celastrol against A549 (IC_50_ = 5969 nM) and HUVECs (IC_50_ = 623.3 nM) by cell counting kit 8 (CCK8) (Appendix A), which illustrated that celastrol showed the stronger cytotoxicity against HUVECs than A549. Consequently, we mainly focused on the effect of celastrol on HUVECs. we found that the cell viability of HUVECs was attenuated under 500 nM celastrol (Figure 2A), while there were no obvious changes under 250 nM celastrol, hence, 250 nM was chosen as the concentration to study the pharmacological effects of celastrol in vitro. The tube formation of HUVECs in Matrigel was suppressed by celastrol compared with its control group (Figure 2B,C), the nodes of tubes were dramatically reduced, which was followed by a decrease in the number of tubes and a decrease in tube length. This result suggests that celastrol may influence tip cells’ function in inducing HUVEC migration by extending filopodia at the forefront of the vascular branch [21], thus inhibiting the following tube formation process. Then, we used Mito-tracker specifically staining mitochondria to quantify the length distribution changes of HUVECs after celastrol treatment. After analysis with the High Connotation Imaging Analysis System, we found that mitochondrial length was markedly reduced in the celastrol-treated group compared with its control group (Figure 2D). There are more mitochondria with lengths of less than 2 nm, while there are fewer mitochondria with lengths of more than 5 nm and less than 12 nm in the celastrol-treated group (Figure 2E). In subsequent experiments, we found that the migration capacity of HUVECs was also decreased under celastrol treatment (Figure 2F), thus attenuating the ability of tube extension (Figure 2B,C). Through a series of in vitro experiments, we confirmed that celastrol can inhibit angiogenesis, which is consistent with the results of in vivo experiments. However, which molecule or signaling pathway is responsible for this needs to be further explored.

### 3.3. Celastrol Affects Mitochondrial Function and Morphology

Angiogenesis contains multistage processes, mainly including endothelial cell activation, sprout, migration, proliferation, alignment, tube formation, and branching. Energy supply plays a critical role in these processes, thus, we examined the effects of celastrol on mitochondrial function in HUVECs by the Seahorse XF Mito Stress Test (Figure 3A). The real-time OCRs illustrated that celastrol treatment notably decreased the mitochondrial basal respiration, maximal respiration, spare respiratory capacity, and ATP production (Figure 3B), suggesting a lower cellular respiratory function in HUVECs, which may lead to energy supply deficiency. These results showed that celastrol can significantly affect mitochondrial function. Mitochondria are dynamically changing organelles, their morphological structures changes can lead to functional alternations, and we have verified that the length of mitochondrial was dramatically decreased in the celastrol-treated group (Figure 2D,E), suggesting that celastrol may change mitochondrial function by promoting mitochondrial fission. Thus, a Western blot assay was carried out to evaluate the expression changes of mitochondrion morphology-related proteins (Figure 3C,D). Celastrol did not affect the expression of OMM fusion proteins MFN 1 and 2, while the expression of TOM20, a mitochondrial transporter protein located in OMM, was not significantly affected by celastrol. The expression of COX Ⅳ, which is regarded as a mitochondrial marker and located in IMM, was dramatically increased under celastrol treatment, coincidentally, the content of IMM fusion protein OPA1 was significantly decreased. By immunofluorescence staining of tumor tissues, the obviously decreasing expression of CD31 was observed in the celastrol group compared with the control group, which represents a lack of blood vessels inside the tumor. The OPA1 expression was consistent with in vitro results, and was also reduced in the presence of celastrol (Figure 3E).

### 3.4. Celastrol Inhibits Angiogenesis by Downregulating OPA1

Analyzing from a mitochondrial perspective, we found that the morphology-related protein OPA1 may be one of the key factors for celastrol to inhibit tumor angiogenesis, thus we constructed OPA1-deficient HUVECs with siRNA to verify this conjecture (Figure 4A,B). According to the Matrigel tube formation assay, OPA1-deficient HUVECs demonstrate a similar correspondence to the celastrol group, which noticeably diminished the nodes of tubes, number of tubes, and tube length (Figure 4C,D). Consistent with the celastrol treatment group, the mitochondrial length of the OPA1-deficient group was much shorter than that of the control group (Figure 4E,F), and the same outcome was shown in the wound-healing migration assay (Figure 4G). Under TEM, fracture and disordered arrangement of cristae in mitochondrial were observed in both the celastrol group and OPA1 siRNA group (Figure 4H). OPA1 has been reported to be a core molecule in tumor angiogenesis [16], and a series of experiments have confirmed that celastrol inhibits angiogenesis mainly by downregulating OPA1 expression, but more specific molecular mechanisms involved need to be further investigated.

### 3.5. Celastrol Inhibits Angiogenesis via STAT3/OPA1/p65 Pathway

OPA1 requires post-translational processing to produce L- and S-OPA1, which is accomplished by the proteases OMA1 and YME1L [12,13,14]. The expression levels of OMA1 and YME1L were not affected by celastrol according to Western blot analysis (Figure 5A,B), suggesting that celastrol may not alter the post-translational processing of OPA1. The mRNA level of OPA1 was noticeably lower in the celastrol group (Figure 5C), indicating that celastrol may alter OPA1 transcription by affecting nuclear transcription factors. By further analysis, celastrol administration resulted in significant downregulation of phosphorylated STAT3 (p-STAT3) in protein levels without marked changes of STAT3 (Figure 5D,E). According to the molecular docking study, celastrol can bind to the Glu 638 of STAT3, which is located within the Scr homology 2 domain (SH2, residual 584 to 688) (Figure 5F). SH2 is essential for STAT3 dimer formation after phosphorylation [22], and, in turn, inhibits STAT3 translocation into the nucleus and the expression of downstream genes. Stattic is an inhibitor of STAT3 and effectively inhibits its phosphorylation at the sites Y705 and S727 [23] (Figure 5H). The viability of HUVECs showed no significant change under 500 nM stattic treatment, while under 1000 nM stattic, its viability was decreased (Figure 5G). Hence, 500 nM was chosen as the concentration to study the pharmacological effects of stattic in vitro. Consistent with that of the celastrol treatment group, the protein level of OPA1 was lower than that of the control group after stattic administration (Figure 5H), and the same outcome was shown in the Matrigel tube formation assay (Figure 5I,J): the nodes of tubes, the number of tubes, and tube length were decreased under celastrol or stattic treatment. The measurement of mitochondrial length illustrated that the mitochondrial lengths of the celastrol group and stattic group were much shorter than that of the control group (Figure 5K,L). In conclusion, the downregulation of p-STAT3 is mainly responsible for the inhibition of OPA1 by celastrol, while the upregulation of p-P65 is the main reason for the inhibition of angiogenesis via OPA1 [16] (Figure 5D,E).

## 4. Discussion

This study first reported that celastrol inhibited lung-derived tumor growth in CDX, and the vessels inside the tumor were dramatically decreased under celastrol treatment. Compared with targeting tumor cells, anti-angiogenesis therapy mainly focuses on endothelial cells (ECs), mural cells, and stromal cells, showing more stabilization and less drug resistance due to genomic stability. The classical anti-angiogenesis therapy was predominately based on the VEGF pathway, FDA has proven to be a humanized neutralizing monoclonal antibody bevacizumab, which can block human VEGF [24,25]. Small molecule sorafenib and sunitinib have also been proven as tyrosine kinase inhibitors (TKIs) of VEGF receptors (VEGFR) [26]. This kind of strategy has been regarded as the most promising therapy since the 1970s, but the clinical facts do not go in the same direction as most preclinical models [6,27]. The heterogeneity and overexpression of growth factors caused by genetic instability of the tumor cells may take responsibility for the clinical results, hence anti-VEGF therapy must combinate with other drugs to gain a beneficial effect and fewer side effects [25,28]; unfortunately, this combination sometimes does not work [29,30]. In addition to its role in anti-angiogenesis, celastrol has also shown good performance in inducing apoptosis and inhibiting metastasis of tumor cells [31], which has demonstrated a comprehensive and effective role in anti-tumor therapy. Therefore, multi-targeted drugs may hold more promise than previous single-targeted drugs.

In this study, we used 250 nM celastrol, which had no obvious effect on the cell viability of HUVECs but had a significant inhibitory effect on tube formation and migration in vitro. In previous reports, 1 and 2 µM celastrol as well as 0.25 to 2 µg/mL (equal to 0.555 to 4.438 µM) celastrol under hypoxia conditions (1% O_2_) can effectively depress tube formation of HUVCEs in matrix gel [4,8], while the same results can be achieved at a lower dosage in our assay. The 50 nM to 2 µM celastrol can obviously suppress the migration of HUVECs in wound-healing assay [4,32]. It is worth noting that 50 nM is five times lower than the dose of celastrol we used, and these results exhibit the excellent migration inhibition capacity of celastrol. Lower dosage requirements in preclinical studies mean fewer side effects occurring in clinical studies. Apart from that, celastrol also has an inhibitory effect on the migration of tumor cells, such as the ccRCC cell line (786-O cell) [2] and colorectal cancer (CRC) cell lines (HT29 and HCT116) [32], and the multiple anti-tumor capabilities further highlight the potential of celastrol.

In studies of the pharmacological mechanisms of celastrol, increasing attention has been paid to its effects on mitochondria. The most significant effect of celastrol on mitochondria is the induction of mitochondria-dependent apoptosis, a process that may begin with drug-induced over-accumulation of reactive oxygen species (ROS), leading to a decrease in mitochondrial membrane potential and modulating the expression or cleavage of apoptosis-associated molecules [1,33]. In this paper, the results of the Seahorse XF Mito Stress Test illustrate that celastrol treatment can significantly decrease mitochondrial basal respiration, maximal respiration, spare respiratory capacity, and ATP production. The decrease in basal respiration represents the declining energetic demand of HUVECs under baseline conditions; the decrease in maximal respiration represents the reduction of mitochondrial maximum operation capacity and respiration rate, combined with the lowering ATP production, showing that celastrol may inhibit angiogenesis by restricting mitochondrion energy supply ability. This study reveals the inhibition of mitochondrial functional capacity by celastrol from the perspective of energy metabolism and enriches the pharmacological effects of this drug.

In addition to energy metabolism changes, we found that celastrol fragmented the mitochondria of HUVECs by using the Operetta CLS High-Content Analysis System. Mitochondria maintain their functional capacity by dynamically regulating fission and fusion, and their length changes may also reflect the working capacity to some extent. Based on this phenomenon, we found by further analysis that celastrol significantly reversed VEGF-induced OPA1 upregulation. OPA1 is a key molecule located on the IMM that mediates IMM fusion, so we speculated that the downregulation of OPA1 caused by celastrol may be the main reason for the mitochondrial fragmentation of HUVECs. The previous reports demonstrate that OPA1 plays an important role in tumor angiogenesis [16], which provides stronger evidence to validate the promising capacity of celastrol in anti-tumor angiogenesis therapy. OPA1 can be cleavaged by MMP into L-OPA1 and S-OPA1, both are essential for mediating mitochondrial fusion, and the fusion efficiency is highest when the ratio is 1:1 [15]. However, in this study, celastrol showed no influence on OPA1 cleavage, in the meantime, the expression levels of their major hydrolases OMA1 and YME1L were not significantly changed. Interestingly, nuclear transcription factor p-STAT3 was significantly downregulated under celastrol administration. Meanwhile, stattic (STAT3 inhibitor) exhibited the same OPA1 downregulation, angiogenesis inhibition, and mitochondrial fragmentation effects as celastrol. These confirmed that celastrol may suppress angiogenesis by reducing the energy supply of HUVECs through inhibition of STAT3 phosphorylation, which in turn induced OPA1 downregulation and mitochondrial fragmentation. It is worth mentioning that in the Matrigel tube formation assay, there were differences in cell growth rates due to the long interval between several experiments and the different generations of HUVECs; however, the statistical results were similar, and the inhibitory effect of celastrol on the tube formation of HUVECs was more pronounced when their growth rate was faster (Figure 2B,C, Figure 4C,D, and Figure 5J,K), which further verified the anti-angiogenesis effect of celastrol. In a previous study, the upregulation of OPA1 promoted tumor angiogenesis by suppressing NF-κB activation [16], while in this study, celastrol administration decreased OPA1, followed by p-P65 upregulation, which is the signal of NF-κB activation, thus showing the anti-angiogenesis effect.

## 5. Conclusions

As shown in Figure 6, this study first demonstrates that celastrol inhibits the mitochondrial-sharping protein OPA1, which begins with the downregulation of p-STAT3, leading to mitochondrial fragmentation and NF-κB activation, and eventually resulting in suppression of angiogenesis in vivo and in vitro.

## Figures and Tables

**Figure 1 pharmaceutics-15-00048-f001:**
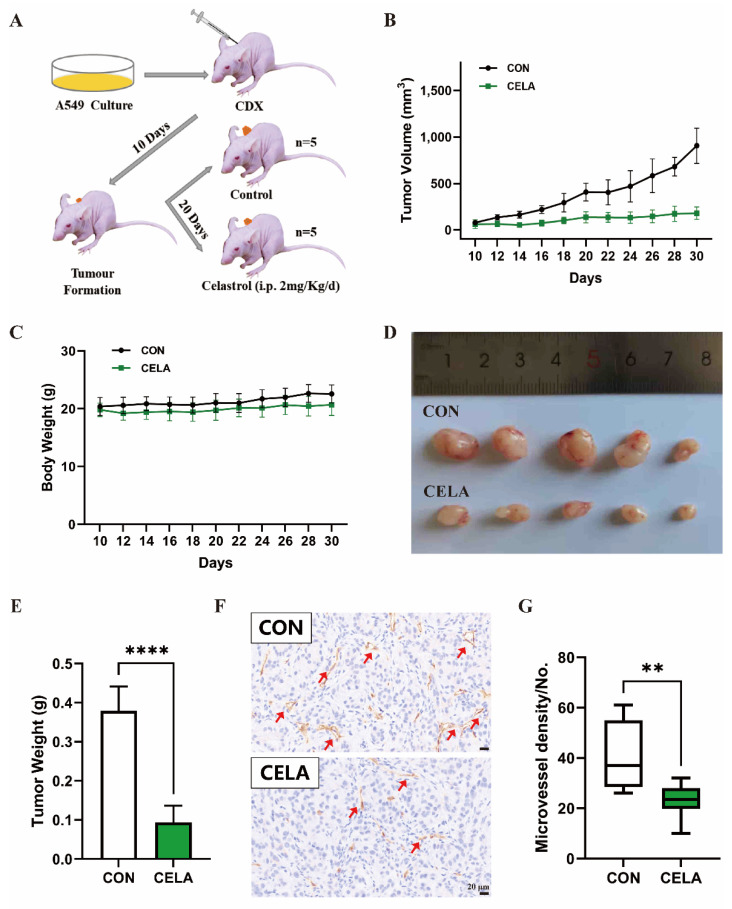
Celastrol showed an inhibitory effect on lung-derived tumor growth and angiogenesis in vivo: (**A**) in vivo experimental design; (**B**) the changes in tumor volume and (**C**) body weight under celastrol treatment; (**D**) macrophotograph of the tumors; (**E**) the effects of celastrol on tumor weight; (**F**,**G**) microvessel density change showed by H&E staining. Red arrows indicate blood vessels in tumor tissue. ** *p* < 0.01, **** *p* < 0.0001 vs. the control group.

**Figure 2 pharmaceutics-15-00048-f002:**
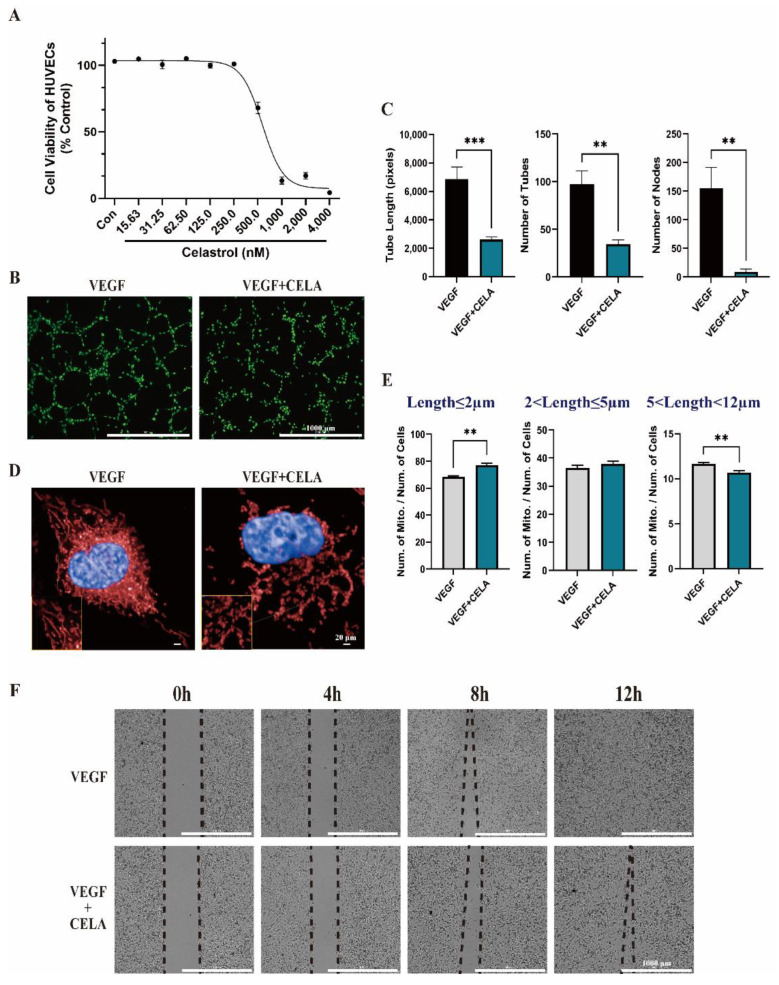
Celastrol showed an inhibitory effect on tumor angiogenesis in HUVECs. (**A**) Cell viability of HUVECs after 24 h celastrol treatment. (**B**) The effects of celastrol on Matrigel tube formation. (**C**) The effects of celastrol on the nodes of tubes, the number of tubes, and tube length. (**D**,**E**) The effects of celastrol on mitochondrial length evaluated by high-content screening. (**F**) The wound-healing migration assay showed the effects of celastrol on the migration capacity of HUVECs. ** *p* < 0.01, *** *p* < 0.001.

**Figure 3 pharmaceutics-15-00048-f003:**
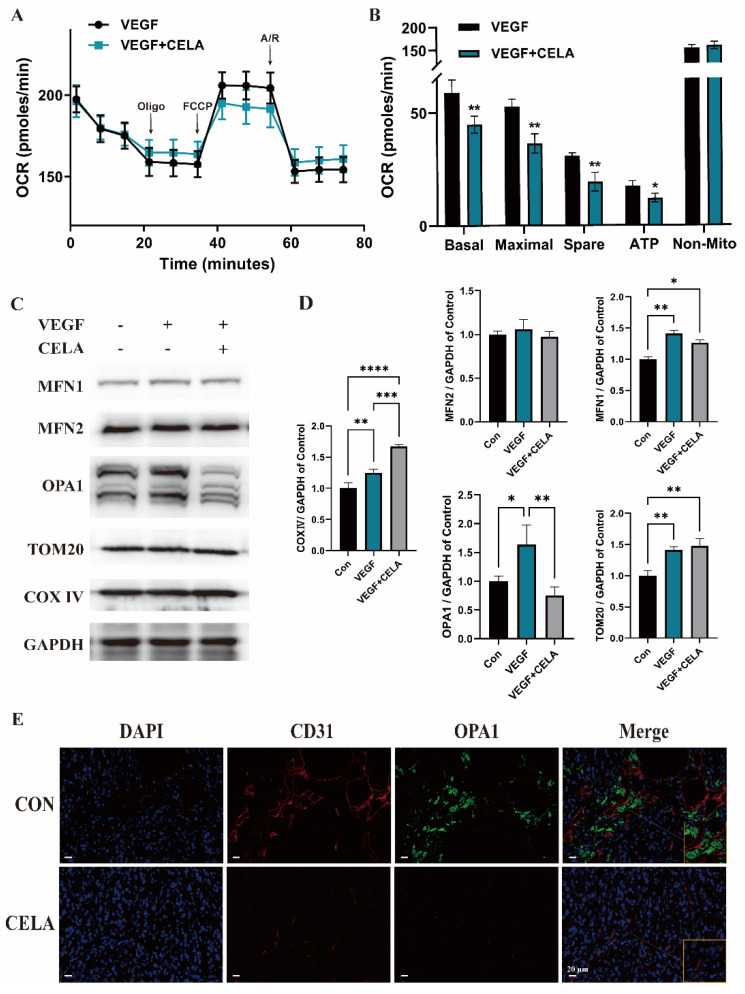
Celastrol affects mitochondrial function and the expression of morphology-related proteins. (**A**) The effects of celastrol on OCR. (**B**) The effects of celastrol on mitochondrial basal respiration, maximal respiration, spare respiratory capacity, and ATP production. (**C**,**D**) The effects of celastrol on protein levels of MFN1, MFN2, OPA1, TOM20, COX IV, and GAPDH. (**E**) The expression of CD31 and OPA1 in tumor tissues. * *p* < 0.05, ** *p* < 0.01, *** *p* < 0.001, **** *p* < 0.0001.

**Figure 4 pharmaceutics-15-00048-f004:**
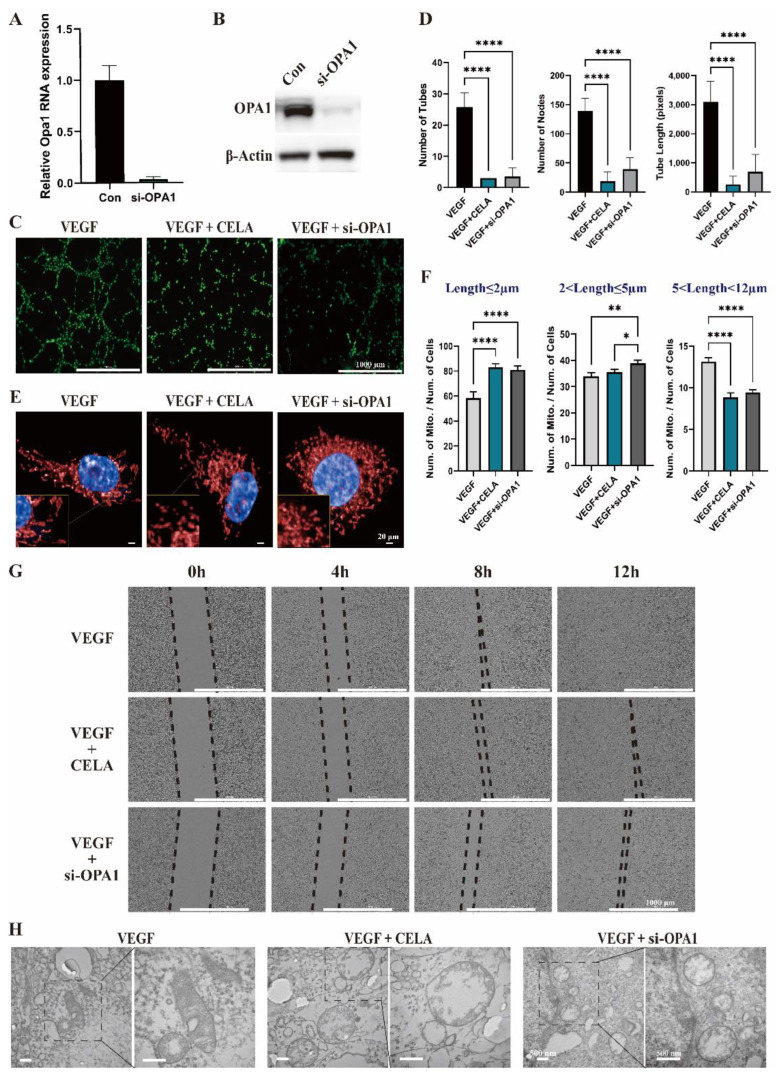
Celastrol inhibits angiogenesis by downregulating mitochondrial fusion protein OPA1. (**A**,**B**) Validation of siRNA knockdown OPA1 by qPCR and Western blot. (**C**) The effects of celastrol on Matrigel tube formation in HUVECs and OPA1 knockdown HUVECs. (**D**) The effects of celastrol on the nodes of tubes, the number of tubes, and tube length in HUVECs and OPA1 knockdown HUVECs. (**E**,**F**) The effects of celastrol on mitochondrial length in HUVECs and OPA1 knockdown HUVECs evaluated by high-content screening. (**G**) The wound-healing migration assay showed the effects of celastrol on the migration capacity of HUVECs and OPA1 knockdown HUVECs. (**H**) The changes of mitochondrial structure of HUVECs in the celastrol group and OPA1 siRNA group under TEM. * *p* < 0.05, ** *p* < 0.01, **** *p* < 0.0001.

**Figure 5 pharmaceutics-15-00048-f005:**
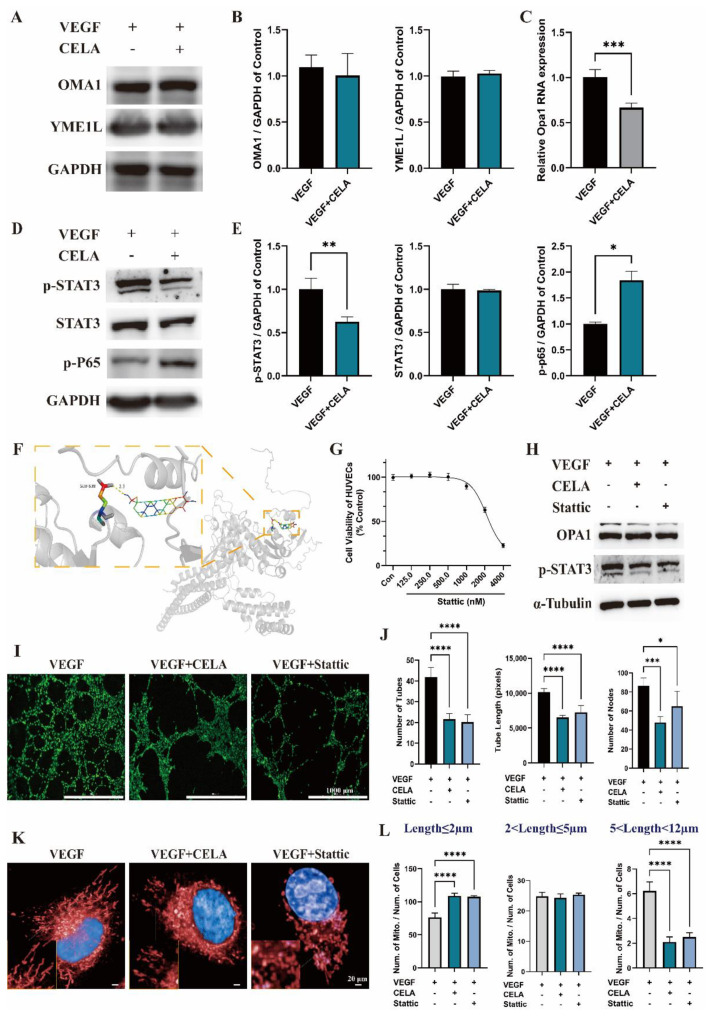
Celastrol downregulates OPA1 by triggering the phosphorylation of STAT3. (**A**,**B**) The effects of celastrol on protein levels of OMA1 and YME1L. (**C**) The effects of celastrol on OPA1 mRNA. (**D**,**E**) The effects of celastrol on protein levels of p-STAT3, STAT3, p-P65, and GAPDH. (**F**) Schematic diagram of binding between celastrol and STAT3 protein, protein is shown in the gray cartoon, the ligand is represented by rainbow sticks. (**G**) Cell viability of HUVECs after 24 h stattic treatment. (**H**) The effects of stattic on protein levels of OPA1, p-STAT3, and α-Tubulin. (**I**,**J**) The effects of stattic on the nodes of tubes, the number of tubes, and tube length in HUVECs. (**K**,**L**) The effects of stattic on mitochondrial length in HUVECs evaluated by high-content screening. * *p* < 0.05, ** *p* < 0.01, *** *p* < 0.001, **** *p* < 0.0001.

**Figure 6 pharmaceutics-15-00048-f006:**
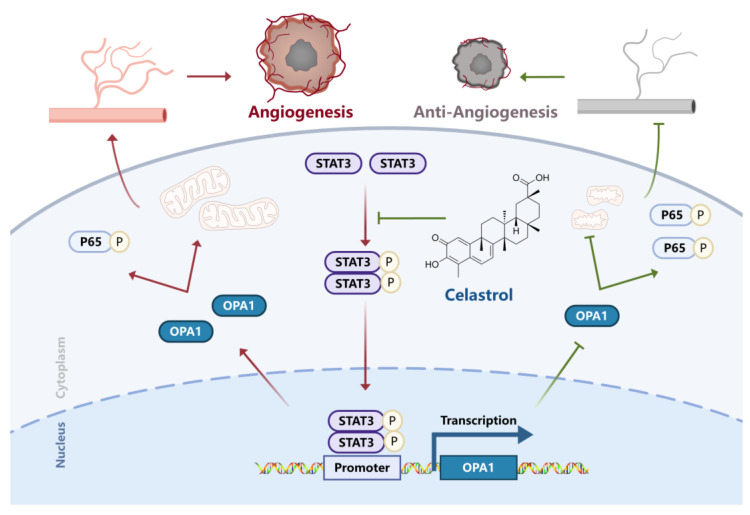
Schematic representation of celastrol inhibiting angiogenesis.

## Data Availability

The data presented in this study are not publicly available due to privacy restrictions, please request from the corresponding author if needed.

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
