# Peer review of "Targeting OPA1-Mediated Mitochondrial Fusion Contributed to Celastrol’s Anti-Tumor Angiogenesis Effect"

_pharmaceutics, 2022, doi:10.3390/pharmaceutics15010048_

Round 1

Reviewer 1 Report

The manuscript entitled “Targeting Opa1-mediated Mitochondrial Fusion Contributed to Celastrol’s Anti-tumor Angiogenesis Effect” reports that celastrol induces angiogenesis inhibition in NSCLC and decreases the mitochondrial length and function of HUVECs cells. This is a well-designed study, but the Material and Methods section needs to be improved.

1. In section 2.2, the number of inoculated cells per mouse needs to be indicated. The amount of celastrol administrated is missing (i.p. ? mg/g/day) (line 91). Additionally, the Authors say that they “kept feeding for 20 days” (line 91), but in the results section, it’s said that “After 30 days’ feeding” (line 202). Please clarify.

2. In line 107, the 250 nM concentration is missing.

3. The cell density used in the Matrigel tube formation assay and mitochondrial length measurement need to be included in the description.

4. Figure 6 should be improved to better communicate the molecular mechanisms of angiogenesis inhibition.

5. A few typos can be found in the manuscript (e.g., “Mearment” instead of measurement, line 124). Please revise the whole manuscript.

Reviewer 2 Report

-          For the in vivo study, it would be better to compare the results with a standard drug to indicate the therapeutic effect of the tested compound.

-          What about to apply the molecular docking study for celastrol with the effective target.

-          How did the authors orient the mechanism of cell death to angiogenesis rather than apoptosis, as the STAT3 was also included in the apoptotic pathway.

-          For the western blotting images, authors should provide the image for the full image.

-          Authors should provide the scale bar for spectroscopic images with adding arrows for illustration.

-          Authors should calculate the IC50 for the in vitro assay, and this may represent the cytotoxicity of celastrol against A549. 

Round 2

Reviewer 2 Report

The authors have addressed all comments, and the manuscript can be accepted in its current form. Please attach all figures raised by reviewer comments in the supplementary file